# Land Use/Land Cover Mapping Using Multitemporal Sentinel-2 Imagery and Four Classification Methods—A Case Study from Dak Nong, Vietnam

**Huong Thi Thanh Nguyen [1], Trung Minh Doan [1], Erkki Tomppo [2,3,\*] and Ronald E. McRoberts [4]**

[1]  Department of Forest resource & Environment management (Frem), Faculty of Agriculture and Forestry, Tay Nguyen University, Le Duan Str. 567, Buon Ma Thuot City 63000, Daklak Province, Vietnam; nguyenthithanhhuong@ttn.edu.vn (H.T.T.N.); doantrung@ttn.edu.vn (T.M.D.)

[2]  Department of Electronics and Nanoengineering, School of Electrical Engineering, Aalto University, P.O. Box 11000, 00076 Aalto, Finland

[3]  Department of Forest Sciences, University of Helsinki, Latokartanonkaari 7, P.O. Box 27 FI-00014 Helsinki, Finland

[4]  Raspberry Ridge Analytics, 15111 Elmcrest Avenue North, Hugo, MN 55038, USA; mcrob001@umn.edu

\*  Correspondence: erkki.tomppo@aalto.fi

**Abstract:** Information on land use and land cover (LULC) including forest cover is important for the development of strategies for land planning and management. Satellite remotely sensed data of varying resolutions have been an unmatched source of such information that can be used to produce estimates with a greater degree of confidence than traditional inventory estimates. However, use of these data has always been a challenge in tropical regions owing to the complexity of the biophysical environment, clouds, and haze, and atmospheric moisture content, all of which impede accurate LULC classification. We tested a parametric classifier (logistic regression) and three non-parametric machine learning classifiers (improved k-nearest neighbors, random forests, and support vector machine) for classification of multi-temporal Sentinel 2 satellite imagery into LULC categories in Dak Nong province, Vietnam. A total of 446 images, 235 from the year 2017 and 211 from the year 2018, were pre-processed to gain high quality images for mapping LULC in the 6516 km$^2$ study area. The Sentinel 2 images were tested and classified separately for four temporal periods: (i) dry season, (ii) rainy season, (iii) the entirety of the year 2017, and (iv) the combination of dry and rainy seasons. Eleven different LULC classes were discriminated of which five were forest classes. For each combination of temporal image set and classifier, a confusion matrix was constructed using independent reference data and pixel classifications, and the area on the ground of each class was estimated. For overall temporal periods and classifiers, overall accuracy ranged from 63.9% to 80.3%, and the Kappa coefficient ranged from 0.611 to 0.813. Area estimates for individual classes ranged from 70 km$^2$ (1% of the study area) to 2200 km$^2$ (34% of the study area) with greater uncertainties for smaller classes.

**Keywords:** classification; Sentinel 2; land use land cover; improved k-NN; logistic regression; random forest; support vector machine

## 1. Introduction

### 1.1. Motivation

Most Vietnamese forests are classified as tropical with natural forest accounting for more than 70% of the total forest area [1]. Dak Nong province has the most abundant natural forest resources in

Vietnam. The great diversity of this resource is primarily owing to a wide variety of environmental and climatic factors, most of which are governed by latitude and topography [2]. However, Dak Nong's natural forests are being lost at an alarming rate owing to factors that include expanding agriculture, conversion to commercial and plantation forest types, and increasing human population. For many years, the Highland Plateau, which includes Dak Nong, has been a major "hot spot" for conversion of forest to agriculture in Vietnam. During the 1990s and early 2000s, forest was lost at an average annual rate of 15,000 ha per year [3], with forest cover declining from 75% in 1985 to 60% in 2009. During this time, the annual rate of deforestation in the Highland Plateau was the greatest of all regions, accounting for 46.3% of the entire national forest area lost.

The Highland Plateau is characterized by a complex topography with mountains, highlands, valleys, deltas, and diversified soil types. Approximately 1.3 million ha are fertile soils, rich in organic matter and nutrients, that facilitate development of high value industrial perennial crops such as coffee, rubber, pepper, cashew, and fruit trees. Additionally, the distinct rainy and dry seasons in the south of Vietnam cause differences in the rates of plant growth. Finally, climatic differences from north to south in Vietnam cause vegetation to vary in physiognomy and lead to morphological differences among land cover types, particularly between semi-evergreen and deciduous Dipterocarp forests.

Current, accurate, and detailed land cover information that reflects these unique topographic and climatic conditions, particularly for natural forest types, is crucial for land managers, decision makers, and policy makers tasked with developing forest management strategies and policies [4–6]. Forest resource decision-making is characterized by a large degree of uncertainty regarding the outcomes of alternative choices. The result is a wide variety of opinions regarding the different options that impedes agreement on a clear way forward. Although there is usually agreement on general objectives such as sustainable forest use, biodiversity conservation, and the alleviation of rural poverty, conflicts among stakeholders over the best course of action for achieving these objectives almost always arise. New issues or new actors may appear and influence discussions, external events may unexpectedly require the revision of agreed policy proposals, and deadlocks can exist for long periods, all continuing until pressing circumstances lead to settlements and decisions [7].

### 1.2. Remotely Sensed Data

Remote sensing offers a unique environmental capability for monitoring extensive geographical areas in a cost-efficient manner, while simultaneously producing information related to the Earth's land, atmosphere, and oceans [8]. Land cover mapping represents one of the most common uses of remotely sensed data [9–11], with satellite imagery serving as one of the most important data sources [11].

As previously, Dak Nong presents unique challenges for the construction of accurate remote sensing-based land use land cover (LULC) maps [12]. The variation in vegetation owing to the rainy/dry seasonal variation affects the spectral reflectance properties of vegetation. For example, deciduous dipterocarp forests have spectral properties in the dry season that are similar to those of other cover types such as industrial coffee and rubber crops, whereas the respective spectral properties are quite different in the rainy season. Only a few studies have accommodated this kind of seasonal variation when constructing satellite image-based land cover classifications. Sothe et al. (2017) [13] combined multi-spectral fall and spring season images when mapping land cover with Landsat-8 data and found that the inclusion of additional band data considerably improved classifications when compared with the use of fall spectral bands alone. For both classifiers used by Sothe et al. (2017) [13], there were meaningful increases in classification accuracy, by 4.8% and 2.9% for the random forests and support vector machine classifiers, respectively, when the "spring" spectral bands were added.

Dak Nong's seasonal growth variation, varying vegetation spectral signatures, and varying topography suggest that Sentinel-2 satellite spectral data, with its fine spatial resolution (10–60 m), fine temporal resolution (five days), and fine spectral resolution (13 spectral bands), may be particularly well-suited for land cover classification purposes in the province. Although data from the Sentinel-2 sensor have been investigated for a variety of vegetation monitoring [14,15], terrestrial monitoring [16],

and forest classification [16] applications, only a few studies have used Sentinel-2 for land cover mapping [17–19]. Therefore, additional studies that evaluate the utility of this imagery for land cover classification for regions with extremely diverse conditions such as those in Dak Nong are well-justified [20].

*1.3. Classification Techniques*

Factors that affect classification accuracy include sensor type, sources of training and accuracy assessment data, the number of classes, and the classification method [21–23]. Of these factors, the selection of a suitable algorithm that achieves acceptable classification accuracy with minimal processing time can be crucial [24]. Many methods have been proposed for constructing satellite image-based land cover maps [25,26], including both unsupervised and supervised methods and both parametric and non-parametric methods. Although unsupervised algorithms such as IsoData and K-Means clustering have been widely used for many years, general purpose clustering algorithms are cumbersome and difficult to develop [27]. Parametric supervised algorithms such as linear discriminant analysis [28–31] and multinomial logistic regression (MLR) have also been broadly used [32,33] and are often considered standards for comparison purposes [29,30,34]. In the last decade, non-parametric methods including support vector machine (SVM) [35–37], k-nearest neighbors (k-NN) [38,39], and random forests (RF) [40–42] have gained attention for remote sensing-based land cover classification. However, both SVM and RF require the selection of values for multiple parameters that affect their efficacy, and both are computationally intensive [6,35]. For k-NN, Naidoo et al. (2012) [43] reported difficulty in selecting the optimal value of k and that the genetic algorithms recommended for optimization can be computationally intensive [44,45]. Finally, object-based classification has been shown to be an effective method for classifying fine resolution imagery [46,47]. Object-based methods have been used with both fuzzy sets [48,49] and neural networks [48,50] to map land cover using satellite imagery. Although object-based classification methods have been shown to increase accuracy for some land cover mapping applications, fine spatial resolution remotely sensed imagery remains the most frequently used data source for these applications [51].

Because of the unique features of each study and study area including definitions, sample sizes, and numbers and characteristics of the classes, comparisons of methods with respect to accuracy among studies is difficult. Even so, not much effort has been committed to comparing methods with respect to accuracy for diverse tropical forest regions such as Dak Nong. Meyfroidt et al., 2013 [52] used MLR with Landsat data to assess classes of forest change and reported land cover classification accuracies of 0.64 to 0.69. Use of RF for land cover classification has been reported for multiple studies in Vietnam. Bourgoin et al. (in press) [53] used RF with both Landsat and Sentinel-2 data for multiple land cover and land cover change classes and reported an overall accuracy of 0.81. Nguyen et al. (2018) [6] used RF and Landsat data for 10 classes including multiple forest classes in Vietnam. Overall accuracy was approximately 0.90. Ha et al. (2018) [54] used RF and Landsat data for seven land cover classes including forest land and reported overall accuracies greater than 0.90. Finally, Phan and Kappas (2018) [20] reported that SVM was more accurate than RF for classifying six types of land cover types including one forest class in the North of Vietnam (Red River Delta) using Sentinel-2 data. In summary, although only a few studies using only a few methods have been used for the classification of forest land in Vietnam, the reported accuracies are relatively large. Thus, there is merit in a more comprehensive evaluation of classification methods, particularly for diverse tropical regions such as in Dak Nong province, Vietnam, with their distinct seasonal effects.

*1.4. Objectives*

The overall objective was to evaluate the utility of multi-seasonal Sentinel-2 spectral data for land cover classification and mapping in Dak Nong province, Vietnam. A subordinate objective was to compare the parametric MLR and non-parametric ik-NN, SVM, and RF classification methods with respect to both overall and class-level accuracies and with respect to whether the methods exploited

the beneficial effects of the multi-seasonal Sentinel-2 data. Google Earth Engine was used for collecting and pre-processing both training and accuracy assessment data. A second subordinate objective was rigorous statistical estimation of the ground area of each land cover class.

## 2. Materials and Methods

### 2.1. Overview

The structure of this section has multiple components. First, the Dak Nong study area is described in Section 2.2, the Sentinel-2 satellite imagery and its separation into temporal periods are described in Section 2.2, and the land cover data from multiple sources and their separation into training and validation subsets are described in Section 2.3. Next, brief descriptions of the four classifiers are provided in Section 2.4, including descriptions of their statistical properties, details on their required input parameters, and procedures for optimizing their performance. Finally, in Section 2.5, the two analytical components used to compare all combinations of the four temporal image periods and the four classifiers are described. The first component focuses on map accuracy assessment, while the second component focuses on estimating LULC class areas and their corresponding uncertainties. The overall research approach is summarized in Figure 1.

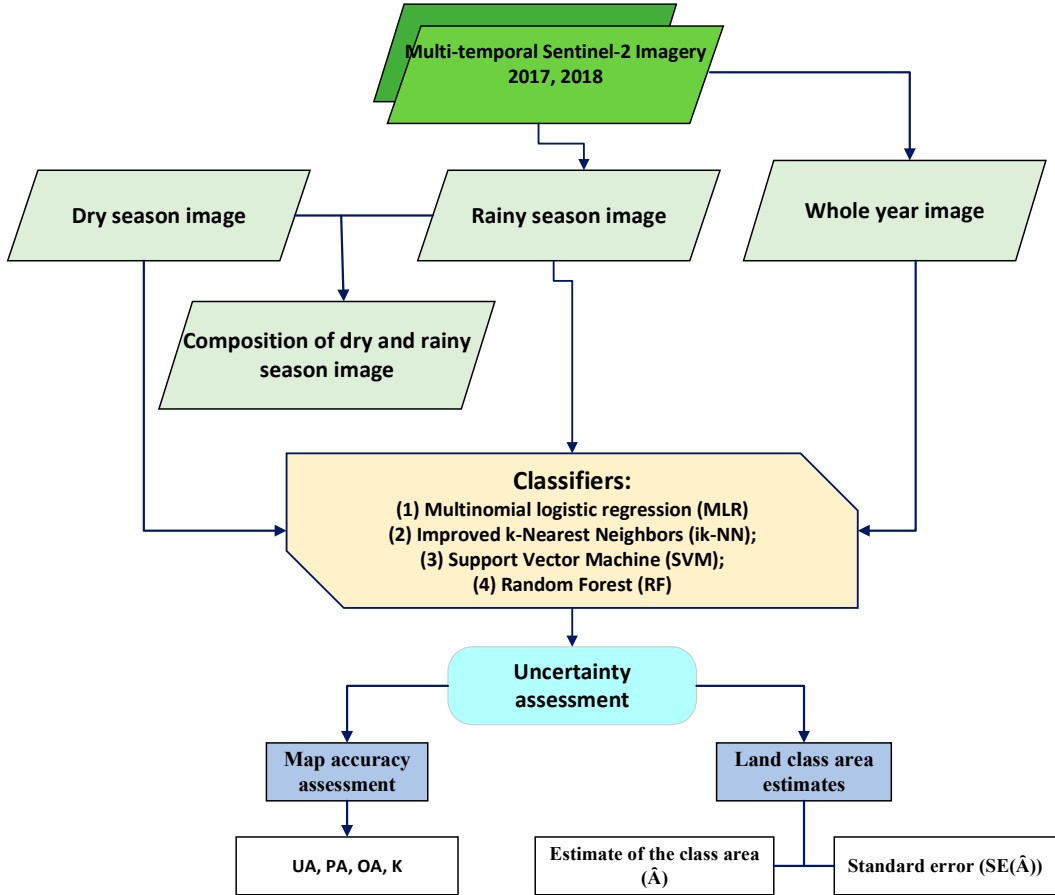

**Figure 1.** Research approach as a flowchart. The Sentinel-2 2017 and 2018 data were collected for different seasons: dry, rainy, whole year, and rainy and dry composited images. The MLR, ik-NN, SVM, and RF classifiers were tested with the resulting uncertainty assessments used as criteria for comparing combinations of seasonal datasets and classifiers. OA, overall accuracy; K, Kappa coefficient; PA, producer's accuracy; UA, user's accuracy.

## 2.2. Study Area

The study was conducted in Dak Nong Province in the Central Highlands of Vietnam (Figure 2). The average elevation is between 600 and 700 m above sea level. The mean temperature is 24 degrees Celsius. The climate conditions produce general characteristics of a subequatorial tropical monsoon climate. The province has characteristics of humid tropical highland climate and is affected by dry-hot southwest monsoons. There are two distinct annual seasons: the rainy season starts in April and ends in November, and the dry season starts in December and ends in March the following year. The average annual rainfall is 2500 mm, of which 90% occurs during the rainy season. The study area extends over 6516 km$^2$ and is characterized by substantial fragmentation, thereby making LULC classification particularly challenging. Natural forest consists of patches of natural evergreen broad-leaved, mixed bamboo, deciduous dipterocarp, and semi-deciduous forest with different levels of disturbance, mainly human in origin.

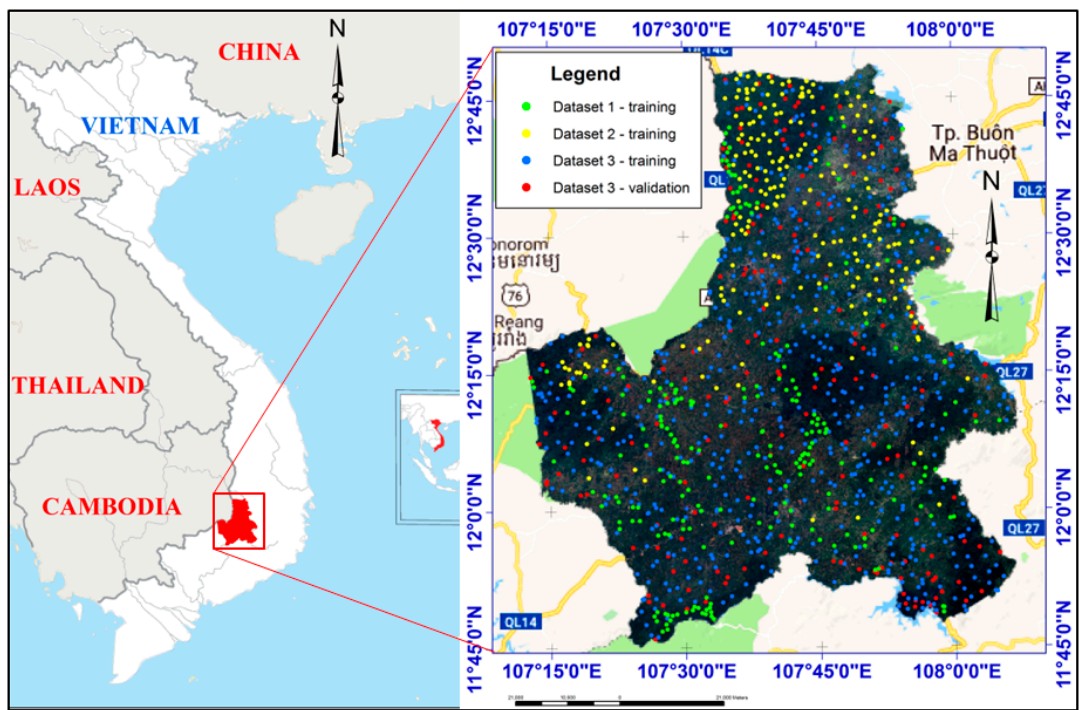

**Figure 2.** The study area in Dak Nong province, Vietnam, with sample unit locations.

## 2.3. Data

### 2.3.1. Sentinel-2 Imagery

Sentinel-2 MSI (multi-spectral instrument, Level-1C) remotely sensed data were used for LULC classification. The Sentinel-2 mission was developed by the European Space Agency (ESA) as a part of the Copernicus Programme [55]. The mission's wide swath, fine spatial resolution (10 m–60 m), multi-spectral features (13 spectral bands), and frequent revisit time (10 days at the equator with one satellite, and 5 days with two satellites) support monitoring vegetation changes within a growing season, forest monitoring, land cover change detection, and natural disaster management [56]. The spectrum characteristics of the Sentinel 2 images are described in Table 1.

**Table 1.** Basic characteristics of Sentinel 2 multi-spectral instrument (MSI).

| Name | Min | Max | Scale | Resolution | Wavelength | Description |
|------|-----|-----|-------|-----------|-----------|-------------|
| B1 | 0 | 10,000 | 0.0001 | 60 Meters | 443 nm | Aerosols |
| B2 | 0 | 10,000 | 0.0001 | 10 Meters | 490 nm | Blue |
| B3 | 0 | 10,000 | 0.0001 | 10 Meters | 560 nm | Green |
| B4 | 0 | 10,000 | 0.0001 | 10 Meters | 665 nm | Red |
| B5 | 0 | 10,000 | 0.0001 | 20 Meters | 705 nm | Red Edge 1 |
| B6 | 0 | 10,000 | 0.0001 | 20 Meters | 740 nm | Red Edge 2 |
| B7 | 0 | 10,000 | 0.0001 | 20 Meters | 783 nm | Red Edge 3 |
| B8 | 0 | 10,000 | 0.0001 | 10 Meters | 842 nm | Near infrared (NIR) |
| B8a | 0 | 10,000 | 0.0001 | 20 Meters | 865 nm | Red Edge 4 |
| B9 | 0 | 10,000 | 0.0001 | 60 Meters | 940 nm | Water vapor |
| B10 | 0 | 10,000 | 0.0001 | 60 Meters | 1375 nm | Cirrus |
| B11 | 0 | 10,000 | 0.0001 | 20 Meters | 1610 nm | Short-wave infrared (SWIR) 1 |
| B12 | 0 | 10,000 | 0.0001 | 20 Meters | 2190 nm | SWIR 2 |
| QA10 | | | | 10 Meters | | Always empty |
| QA20 | | | | 20 Meters | | Always empty |
| QA60 | | | | 60 Meters | | Cloud mask |

The difference in solar illumination geometry during image acquisition between the two seasons was considered in the present study. Although vegetation in the study area presents reduced climatic and phonological seasonality, the observed reflectance varies by season owing to changes in the solar illumination geometry caused by the Earth's translation movement [13]. Therefore, seasonal image datasets were separately classified to evaluate these influences. Accordingly, the scenes of interest included the following: (i) a collection of Sentinel-2 MSI scenes in the study area during the dry season of 2017 and 2018 (1 January 2017 to 31 March 2017, and 1 December 2017 to 31 March 2018), designated imagery 1 (IMG 1) (Table 2); (ii) a collection of Sentinel-2 MSI scenes during the rainy season of 2017 and 2018 (from 1 April 2017 to 30 November 2017 and 1 April 2018 to 30 June 2018), designated IMG 2; (iii) a collection of Sentinel 2 MSI scenes for all months of 2017 (from 1 January 2017 to 31 December 2017), designated IMG 3; and (iv) a combination of all bands for the dry and rainy seasons (combination of IMG 1 and IMG 2, designated IMG 4.

**Table 2.** Seasonal satellite images used in the classification.

| Image Name | Time | Acquisition Date | Number of Images Involved | Number of Bands |
|-----------|------|------------------|---------------------------|-----------------|
| IMG 1 | Dry season, 2017–2018 | 01/01/2017–03/31/2017 and 12/01/2017–03/31/2018 | 169 | 10 |
| IMG 2 | Rainy season, 2017–2018 | 04/01/2017–11/30/2017 and 04/01/2018–06/30/2018 | 277 | 10 |
| IMG 3 | All for year 2017 | 01/01/2017–12/31/2017 | 265 | 10 |
| IMG 4 | Combination of all bands for both 2017 and 2018 (IMG 1 + IMG 2) | Dry season 2017–2018 + Rainy season 2017–2018 | 446 | 20 |

The different seasonal image datasets represented for each season were considered for the analyses based on the median value of the collection. The multi-spectral bands in the study included Blue (B2), Green (B3), Red (B4), Red Edge 1 (B5), Red Edge 2 (B6), Red Edge 3 (B7), near infrared (NIR) (B8), Red Edge 4 (B8A), short-wave infrared (SWIR) 1 (B11), and SWIR 2 (B12). In addition to these spectral bands, the normalized difference vegetation index (NDVI) and a digital elevation model (DEM) were added to the seasonal image data (IMG 1–4) with the objective of increasing classification accuracy, as reported from previous studies [22,57]. These bands, including NDVI [58,59] and DEM, were resampled at 10 m resolution. Image information is described in Table 2 below.

To conduct the analyses, the JavaScript API Code Editor in the Google Earth Engine (GEE) was used to collect data for a large number of images. GEE provides most freely available image data and an application programming interface (API) to analyze and visualize the data [60,61]. Surface reflectance (SR) images for 2017 were not available, and for 2018 images, approximately 50% of the study area was covered by clouds. Hence, a top of atmosphere (TOA) dataset acquired for 2017 and 2018 was used for the study. The set of collected images was pre-processed to reduce the effects of topography and bidirectional reflectance distribution function (BRDF). At the same time, cloud areas were masked out and shadows were removed during this process.

All images underwent pixel-wise cloud and cloud shadow masking using the Google cloudScore algorithm for cloud masking and temporal dark outlier mask (TDOM) for cloud shadows, both of which built on ideas from Landsat TDOM and cloudScore algorithm. The original concept was written by Carson Stam, adapted by Ian Housman, currently documented in [60], and described and evaluated in a forthcoming paper [62]. This study implemented the correction of reflectance spectral values by BRDF based on the method described by Roy et al. 2017a,b [63,64]. Topographic correction is the process to account for diffuse atmospheric irradiance caused by slope, aspect, and elevation effects. The sun-canopy-sensor + C (SCSc) correction method based on the C-correction, as described by Soenen et al. [65], was applied for topographic correction in this study. The median function was then applied to create an image object (single image) representing the median value of all images in the filtered collection [66,67]. The median lies closer to the majority of values and is insensitive to extreme values and has exactly half the values smaller and half the values greater than the median, as elaborately applied by [68]. The post-processed images were then resampled to a spatial resolution of 10 m using the nearest neighbor method [69], and subsetted to the study area. The entire pre-processing was implemented on GEE based on the script available on "Open Geo Blog - Tutorials, Code snippets and examples to handle spatial data" [61,70].

### 2.3.2. Training and Validation Data

Within the study area, 11 LULC classes were distinguished: (1) dense evergreen broadleaved forest (the forest has been slightly impacted); (2) open evergreen broadleaved forest (the forest has been moderately to heavily disturbed); (3) semi-evergreen forest (the forest that consists of a mixture of evergreen and deciduous dipterocarp tree species); (4) deciduous dipterocarp forest; (5) plantation forest; (6) mature rubber (≥3 years old); (7) perennial industrial plants; (8) croplands (annual crop land); (9) residential area; (10) water surface; and (11) other lands including, but not limited to, other types of grassland, shrubs, bare land, and abandoned land.

Acquiring adequate training and validation data is often challenging in tropical regions. Sothe et al., 2017 [13] and Teluguntla et al., 2018 [71] both obtained good results using sample data from a combination of sources including field investigations, very fine spatial resolution Google Earth imagery, current Landsat and Sentinel imagery, and other sources such as maps. A similar approach was used for this study for which three sets of sample data were acquired in 2017 and 2018: (1) field observations for a purposive sample of size 232; (2) visual interpretations of fine and very fine resolution imagery from sources that included Google Earth for a purposive sample size of 214; and (3) visual interpretations of fine and very fine resolution imagery from sources that included Google Earth and Sentinel- 2A imagery for a simple random sample size of 800. For the latter sample dataset, field observations and data from the 2016 Dak Nong Forest Inventory Map were used to clarify and refine interpretations for the LULC classes such as semi-evergreen forest, plantation forest, and some perennial industrial crops that were difficult to distinguish in the imagery.

To obtain the probability sample necessary for validation, a systematic sample of the probability-based third dataset was selected. The plots in the third dataset were first sorted by their east and north coordinates, and then a systematic sample was selected from within each LULC class. For each class, the proportion selected was arbitrary, but was guided by the desire for a minimum sample size of 15, where possible, while yet retaining a sufficient sample size for training purposes.

For the eleven LULC classes, the proportions were, in order, as follows: 0.20, 0.20, 0.50, 0.67, 1.00, 0.50, 0.11, 0.50, 0.67, 0.50, 0.20. Because the third dataset was selected using a simple random sample, and it was systematically subsampled, each subsample can also be considered a simple random sample and, therefore, can be used for validation. The remaining portion of the third dataset was used for training purposes. The result was a sample size of 1036 for training and a sample size of 208 for validation. The number of validation plots by LULC category was considered sufficient and generally complied with the recommendation of Särndal et al. (1992) [72]. A summary of the training and validation datasets is shown in Table 3 with the spatial distribution of sample locations shown in Figure 2.

**Table 3.** Training and validation data.

| Dataset | Use | Land Cover Class | | | | | | | | | | | Total |
|---|---|---|---|---|---|---|---|---|---|---|---|---|---|
| | | **1** | **2** | **3** | **4** | **5** | **6** | **7** | **8** | **9** | **10** | **11** | |
| 1 | Training | 77 | 6 | 15 | 13 | 29 | 34 | 0 | 13 | 32 | 4 | 9 | 232 |
| 2 | Training | 6 | 8 | 52 | 33 | 11 | 14 | 19 | 21 | 20 | 4 | 25 | 213 |
| 3 | Training | 99 | 97 | 22 | 9 | 0 | 17 | 234 | 20 | 8 | 11 | 74 | 591 |
| Total | Training | 182 | 111 | 89 | 55 | 40 | 65 | 253 | 54 | 60 | 19 | 108 | 1036 |
| 3 | Validation | 25 | 25 | 22 | 17 | 7 | 17 | 28 | 20 | 16 | 12 | 19 | 208 |

### 2.4. Classifiers

The MLR, ik-NN, RF, and SVM supervised classification algorithms were used to classify the satellite image data as described above. The training areas for each LULC type were selected based on Google Earth, field data, and prior knowledge, as well as available data. The models were used as supervised classifiers to classify pixels based on their spectral signatures.

#### 2.4.1. Multinomial Logistic Regression (MLR)

With MLR, the probability of class c for the $i^{th}$ plot, c=1,..., C, is estimated as follows:

$$p\left(y_i = c \middle| \mathbf{x}_i\right) = \frac{\exp(\boldsymbol{\beta}_c \cdot \mathbf{x}_i)}{1 - \sum_{m=1}^{C-1} \exp(\boldsymbol{\beta}_m \cdot \mathbf{x}_i)} + \varepsilon_i, \text{ for } c = 1, \ldots, C-1 \tag{1}$$

and

$$p\left(y_i = C \middle| \mathbf{x}_i\right) = \frac{1}{1 - \sum_{m=1}^{C-1} \exp(\boldsymbol{\beta}_m \cdot \mathbf{x}_i)} + \varepsilon_i, \tag{2}$$

where C is the number of the LULC classes, $\mathbf{x}_i$ is the vector of predictor variable observations for the $i^{th}$ population unit, and $\boldsymbol{\beta}_c$ is the vector of regression coefficients associated with LULC class c. The class with the greatest probability is selected as the prediction for the $i^{th}$ population unit. Optimal estimates for $\{\boldsymbol{\beta}_c : c = 1, \ldots, C\}$ can be obtained using any of multiple statistical software packages.

#### 2.4.2. Improved k-Nearest Neighbors (ik-NN)

In the terminology of nearest neighbors techniques, the auxiliary or predictor variables are designated feature variables and the space defined by the feature variables is designated the feature space; the set of sample units for which observations of both response and feature variables are available is designated the reference set; and the set of population units for which predictions of response variables are desired is designated the target set (Chirici et al., 2016) [73]. All population units for both the reference and target sets are assumed to have complete sets of observations for all feature variables.

For the $i^{th}$ target unit, all forms of nearest neighbors algorithms entail selecting the k-nearest or most similar neighbors, $\left\{y_j^i : j = 1, 2, \ldots, k\right\}$, from the reference set with respect to a distance metric, d, formulated as a function of the feature variables. For categorical response variables such as land cover classes, the prediction, $\hat{y}_i$, for the $i^{th}$ target unit is the most heavily weighted class among the k-nearest

neighbors, a weighted median or mode in case of ordinal scale variables, or a mode in the case of nominal variables. Implementation of nearest neighbors techniques requires multiple selections: (i) the distance metric, d, to assess nearness or similarity in the feature space; (ii) a scheme for weighting the predictor variables in the distance metric; (iii) a scheme for weighting individual neighbors when calculating predictions; and (iv) a number, k, of nearest neighbors [73].

For this study, the distance metric was a simplified version of the metric proposed by Tomppo and Halme (2004) [44], as used in the operational Finnish multi-source national forest inventory (MS-NFI),

$$d_{ij} = \sqrt{\sum_{m=1}^{p} w_m^2 \cdot \left(x_{im} - x_{jm}\right)^2}, \tag{3}$$

where i denotes a target unit; j denotes a reference unit; $d_{ij}$ is the distance between the units i and j; m indexes the feature variables; $x_{im}$ and $x_{jm}$ are observations of the $m^{th}$ feature variable for the $i^{th}$ target unit and $j^{th}$ reference unit, respectively; and $w_m$ is a feature variable weight. Neighbor weights are typically formulated as powers, $t \in [0, 2]$, of distances between target and reference units. Often, the necessary selections to implement a nearest neighbor algorithm are made in an arbitrary method, whereas improved k-NN (ik-NN) entails optimized selection of the weights, $w_m$, using a technique such as genetic algorithms [44,45,74]

### 2.4.3. Support Vector Machine (SVM)

The principle behind the SVM classifier is a hyperplane that separates the data for different classes. The main focus is construction of the hyperplane by maximizing the distance from the hyperplane to the nearest data point of either class. These nearest data points are known as support vectors [75].

According to Huang et al. (2002) [35] (p. 734), by mapping the input data into a high-dimensional space, the kernel function converts non-linear boundaries in the original data space into linear boundaries in the high-dimensional space, which can then be located using an optimization algorithm. Therefore, selection of the kernel function and appropriate values for corresponding kernel parameters, referred to as the kernel configuration, can affect the performance of the SVM.

The radial basis function kernel (RBF kernel) is one of the most popular kernels used to implement the support vector machine algorithm and was used for this study. The squared Euclidean distance metric was used to construct completely non-linear hyperplanes. The RBF kernel of the SVM classifier is commonly used and has performed well. Polynomial kernels, especially high-order kernels, took far more time to train than RBF kernels [35].

Meyer et al. (2002) [76] stated that, for classification tasks, C-classification is most likely used with the RBF kernel because of its good general performance and the small number of parameters (only two, C and $\gamma$). Therefore, the two parameters that must be defined for this classification algorithm are the cost parameter (C) and the kernel width parameter ($\gamma$). According to Knorn et al. (2009) [77] (p.960), C is a regularization parameter that controls the trade-off between maximizing the margin and minimizing the training error. C is a preset penalty value for misclassification errors, while $\gamma$ describes the kernel width, which affects the smoothing of the shape of the class-dividing hyperplane.

The authors of LIBSVM suggest trying small and large values for C, such as 1 to 1000, then using cross-validation to decide which is optimal for the data, and finally trying several $\gamma$'s for the optimal C's. A small C-value tends to emphasize the margin while ignoring the outliers in the training data, while a large C-value may overfit the training data [77] (p.960). Optimal selection of C and $\gamma$ parameters was done by testing C parameters in the range $2^{-1}$ to $2^8$ and $\gamma$ parameters in the range 0.1 to 2.0.

### 2.4.4. Random Forests (RF)

The RF classifier developed by Breiman (2001) [78] requires selection of three parameters: ntree (number of trees to grow), mtry (the number of variables to split each node), and variable importance

(the number of variables/bands influences model performance). Liaw & Wiener (2002) [79] recommend using the square root of the number of input variables as the default value for mtry. A large value for ntree produces a stable result for variable importance, which is estimated using two indicators: (i) mean decrease accuracy (MDA) and ii) mean decrease gini (MDG). MDA is the accuracy associated with each predictor variable based on the out-of-bag error rate (OOB). Gini impurity is a measure of how often a randomly chosen element from the set would be incorrectly labeled if it is randomly labeled according to the distribution of labels in the subset. For this study, MDA values were investigated to determine the importance of variables. Nguyen et al. (2018) [6] indicated that within the range $1 \leq ntree \leq 500$, ntree = 300 produced the best fit. In addition, Breiman (2001) [78] stated that using more than the required number of trees may be unnecessary, albeit not harmful, because the relationship between accuracy and ntree is asymptotic. The 'RandomForest' package in the R environment developed by Liaw and Wiener (version 4.6–14 in 2018) was used in present study. Optimal values of mtry, ntree, and variable importance were selected based on the smallest OOB error. The optimal variable importance depended on the MDA value and accuracy of the model.

### 2.5. Analyses

#### 2.5.1. Accuracy Assessment

Accuracy assessment is an important step before accepting a classification result [21]. The classification accuracy of a map product is estimated by constructing a confusion matrix between reference and classified pixels. Classification accuracy was assessed using criteria such as overall accuracy (OA), Kappa coefficient (K), producer's accuracy (PA), and user's accuracy (UA). Congalton and Green (1999) [80] assert that analysis of the causes of differences in the confusion matrix can be one of the most important and interesting steps in the construction of a map from remotely sensed data.

The objectives of the study included comparing the performance of classifiers as well as assessing the effects of Sentinel-2 satellite images for different seasons, as described in Table 2. The number of seasonal bands used with the four classifiers is reported in Table 4.

**Table 4.** Classifiers and seasonal bands. Ik-NN, improved k-nearest neighbors; MLR, multinomial logistic regression; SVM, support vector machine; RF, random forests.

| Classification Algorithm | Image Set | Number of Bands |
|---|---|---|
| ik-NN | IMG 1 | 10 |
|  | IMG 2 | 10 |
|  | IMG 3 | 10 |
|  | IMG 4 | 20 |
| MLR | IMG 1 | 10 |
|  | IMG 2 | 10 |
|  | IMG 3 | 10 |
|  | IMG 4 | 20 |
| SVM | IMG 1 | 10 |
|  | IMG 2 | 10 |
|  | IMG 3 | 10 |
|  | IMG 4 | 20 |
| RF | IMG 1 | 10 |
|  | IMG 2 | 10 |
|  | IMG 3 | 10 |
|  | IMG 4 | 20 |

#### 2.5.2. Land Cover Class Area Estimation

For each land cover class, an estimate of the class area and the corresponding standard error were calculated using a combination of confusion matrices and stratified estimators [81,82]. For each class,

C, a confusion matrix was constructed for two classes: (i) class C and (ii) the aggregation of all other classes into a single class designated ~C (Table 5). Using estimates of proportions and corresponding variances as indicated in Table 5, the stratified estimate of the area of class C was as follows:

$$\hat{A}_C = A_{tot} \cdot \left( wt_1 \cdot \hat{p}_1 + wt_2 \cdot \hat{p}_2 \right),$$

(4)

with standard error,

$$SE\left(\hat{A}_C\right) = A_{tot} \cdot \left[ wt_1^2 \cdot \hat{Var}\left(\hat{p}_1\right) + wt_2^2 \cdot \hat{Var}\left(\hat{p}_2\right) \right]$$

(5)

where $wt_1$ is the proportion of the total map area in class C, $wt_2 = 1 - wt_1$, and $A_{tot}$ is the total area of interest. Approximate 95% confidence intervals for the class areas can be estimated as follows:

$$\hat{A}_C \pm 2 \cdot SE\left(\hat{A}_C\right).$$

(6)

**Table 5.** Confusion matrix.

| Map Class | Reference Class | | Total | UA * | $\hat{p}_h$ | $\hat{Var}(\hat{p}_h)$ |
|---|---|---|---|---|---|---|
| | C | ~C | | | | |
| C | $n_{11}$ | $n_{12}$ | $n_{1.} = n_{11} + n_{12}$ | $ua_1 = \frac{n_{11}}{n_{1.}}$ | $\hat{p}_1 = \frac{n_{11}}{n_{1.}}$ | $\hat{Var}(\hat{p}_1) = \frac{\hat{p}_1 \cdot (1 - \hat{p}_1)}{n_{1.}}$ |
| ~C | $n_{21}$ | $n_{22}$ | $n_{2.} = n_{21} + n_{22}$ | $ua_2 = \frac{n_{22}}{n_{2.}}$ | $\hat{p}_2 = \frac{n_{21}}{n_{2.}}$ | $\hat{Var}(\hat{p}_2) = \frac{\hat{p}_2 \cdot (1 - \hat{p}_2)}{n_{2.}}$ |
| Total | $n_{.1} = n_{11} + n_{21}$ | $n_{.2} = n_{12} + n_{22}$ | | | | |
| PA * | $pa_1 = \frac{n_{11}}{n_{.1}}$ | $pa_2 = \frac{n_{22}}{n_{.2}}$ | | | | |

* UA = user's accuracy, * PA = producer's accuracy.

## 3. Results

### 3.1. Classifiers

#### 3.1.1. Multinomial Logistic Regression (MLR)

The parameters of the multinomial logistic regression model (Equations (1) and (2)) were estimated using the *multinom* function of the R packages [83]. All variables (spectral values of all bands of the image set in the analysis) were included in the model. The log-likelihood stabilized after 100 iterations. The importance of the variables was quite similar among different Sentinel-2 datasets. For the dry season image (IMG 1) and for the all-month image (IMG 3), the most important variables were Blue and SWIR 2, otherwise, the variable importance values were approximately the same. For the rainy season image (IMG 2), the results were also similar, although the Blue and SWIR 2 importance values were slightly less than for IMG 1 and IMG 3. Differences among importance values were small for the two-season image (IMG 4).

#### 3.1.2. Improved k-NN (ik-NN)

The improved k-NN (ik-NN) algorithm was applied as described in [45], except that only overall accuracy was used in the fitness function. The value of k = 10 was used. For the genetic algorithm, the number of the generations was 60, the population and medi-population sizes were 50, and the maximum number of the random iterations was 4000. Otherwise, the genetic algorithm parameters were as reported by Tomppo et al. (2009) [45]. Because genetic algorithms as a heuristic optimization method may select a local optimum, several trial runs were used to find a near optimal solution. Pixel-level estimates can be readily calculated with ik-NN when the weights of the variables have been optimized. The importance for the different variables was similar for ik-NN and MLR. However, in the case of predictor variables with large correlations, caution should be used when drawing conclusions with either method.

### 3.1.3. Support Vector Machine (SVM)

With the SVM algorithm using the RBF kernel, determination of the optimal cost (C) and Gamma parameters for the model is important. Following Qian et al. (2015) [84] and using our actual dataset, the R function 'tune()' was used to select these two SVM parameters. The optimal cost (C) value was determined from the values: $2^{-1}, 2^0, 2^1, 2^2, 2^3, 2^4, 2^5, 2^6, 2^7, 2^8$, and the Gamma ($\gamma$) value was a free parameter set from 0.1 to 2. The optimal parameters were determined based on classification error. Figure 3 describes the performance of the SVM model using the different cost and Gamma parameters. The darker the blue area, the better the performance of the model presented.

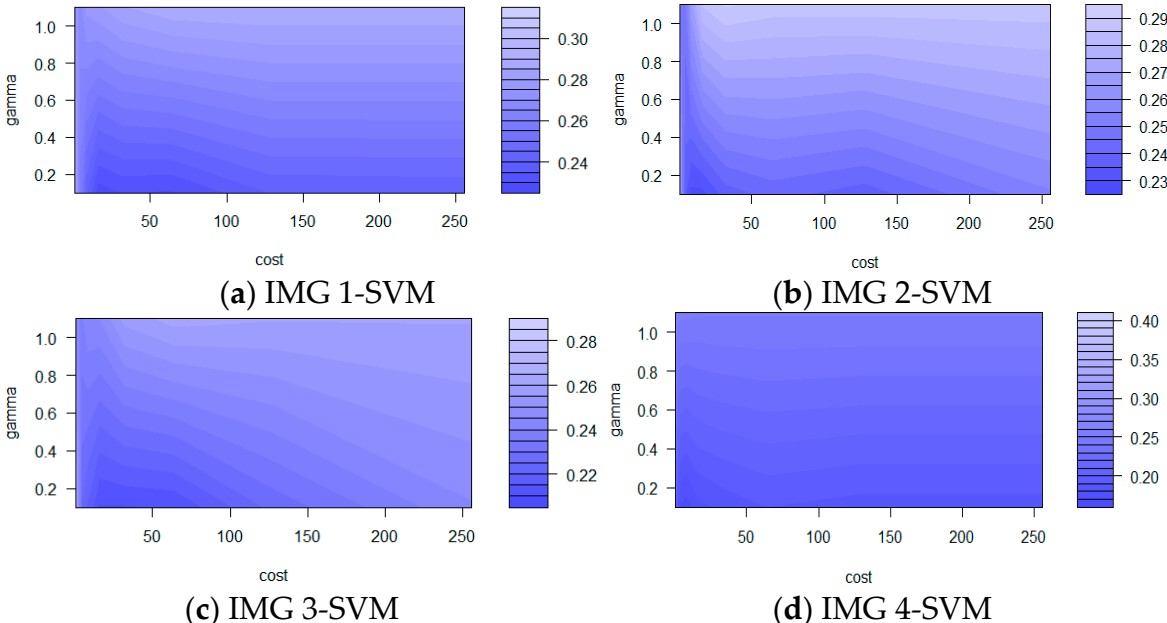

**Figure 3.** Parameter tuning of support vector machine: (**a**) IMG1-SVM; (**b**) IMG 2-SVM; (**c**) IMG 3-SVM; (**d**) IMG 4-SVM.

For the IMG 2-SVM combination and the IMG 4-SVM combination, the optimal C of $2^3$ and Gamma of 0.1 produced classification errors of 0.2283 and 0.1525, respectively, while for the IMG 1-SVM and IMG 3-SVM combinations, the optimal C of $2^5$ and $2^6$, both with Gamma of 0.1, produced classification errors of 0.2212 and 0.2145, respectively.

### 3.1.4. Random Forests (RF)

The three RF parameters, ntree, mtry, and variable importance, play important roles in classification. The algorithm assesses the importance of each variable in the classification process by means of a specific measure. 'Importance()' and 'varImplot()' functions were used to determine the MDA values and to select the potential variables that were actually needed for the optimal RF models. Figure 4 shows the variable importance ranked in the direction of decreasing MDA from right to left for the four seasonal images. The selection of variables was based on MDA using the backward selection method [85]. With this method, the algorithm starts with all predictor variables and then sequentially removes some variables until the greatest accuracy is achieved. Accordingly, the least MDAs were attributed to two bands of IMG 1, 2, and 3: specifically, B6 and B8 for IMG 1 and B6 and B7 for both IMG 2 and IMG 3. For IMG 4, the five bands were included: B5a, B2a, B2b, B7a, and B8b. In addition, the NIR band reduced the accuracy for all images.

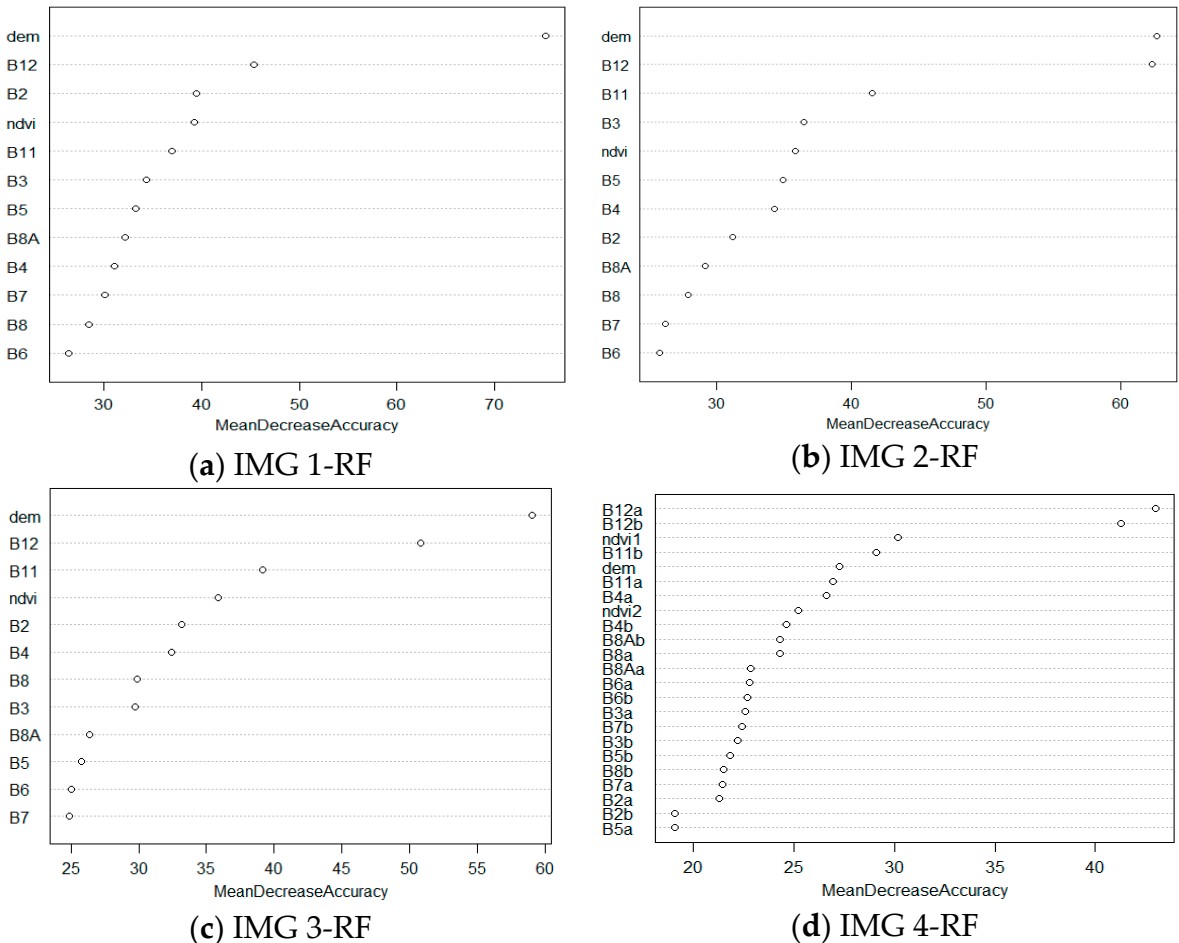

**Figure 4.** Ranking the variable importance measurement (bands): (**a**) IMG 1-RF; (**b**) IMG 2-RF; (**c**) IMG 3-RF; (**d**) IMG 4-RF.

The number of variables used for splitting at each node (mtry) was determined using the tuneRF function based on the variable importance and the number of trees (ntree). On the basis of the OOB error estimation, the optimum ntree and mtry parameters were chosen for the models.

Figure 5 shows the OOB errors when the model was run with ntree ranging from 1 to 500 trees. The smallest OOB errors were obtained for ntree = 500 trees for all seasonal image combinations.

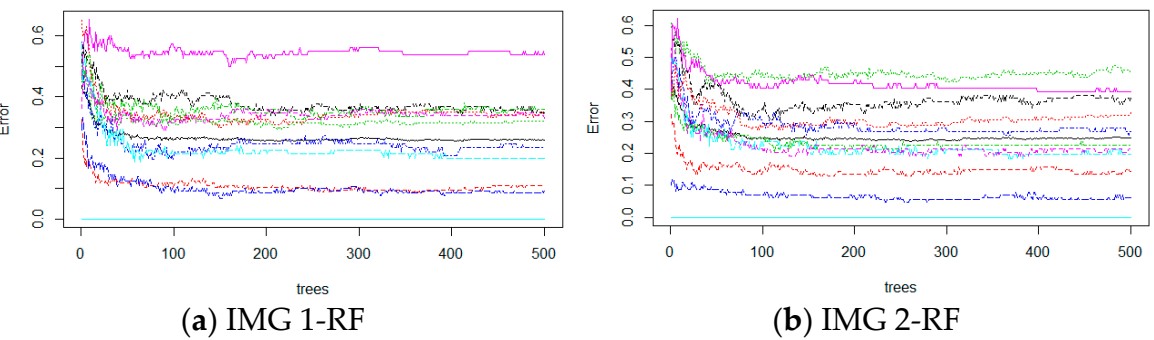

**Figure 5.** *Cont.*

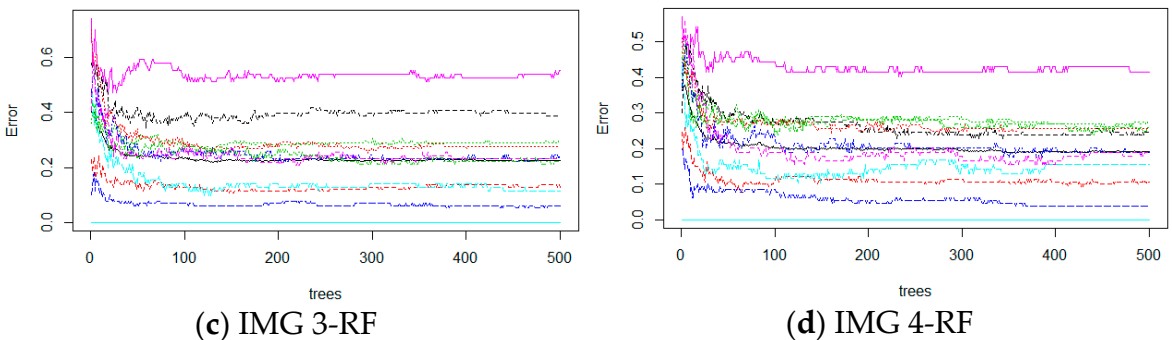

**Figure 5.** Out-of-bag error rate (OOB) errors versus ntree values: (**a**) IMG 1-RF; (**b**) IMG 2-RF; (**c**) IMG 3-RF; (**d**) IMG 4-RF.

The OOB errors associated with different mtry values are shown in Figure 6. The smallest OOB error was obtained with mtry = 3 for the IMG 2/RF and IMG 3/RF combinations, with mtry = 6 for the IMG 1/RF combination, and with mtry = 4 for the IMG4/RF combination.

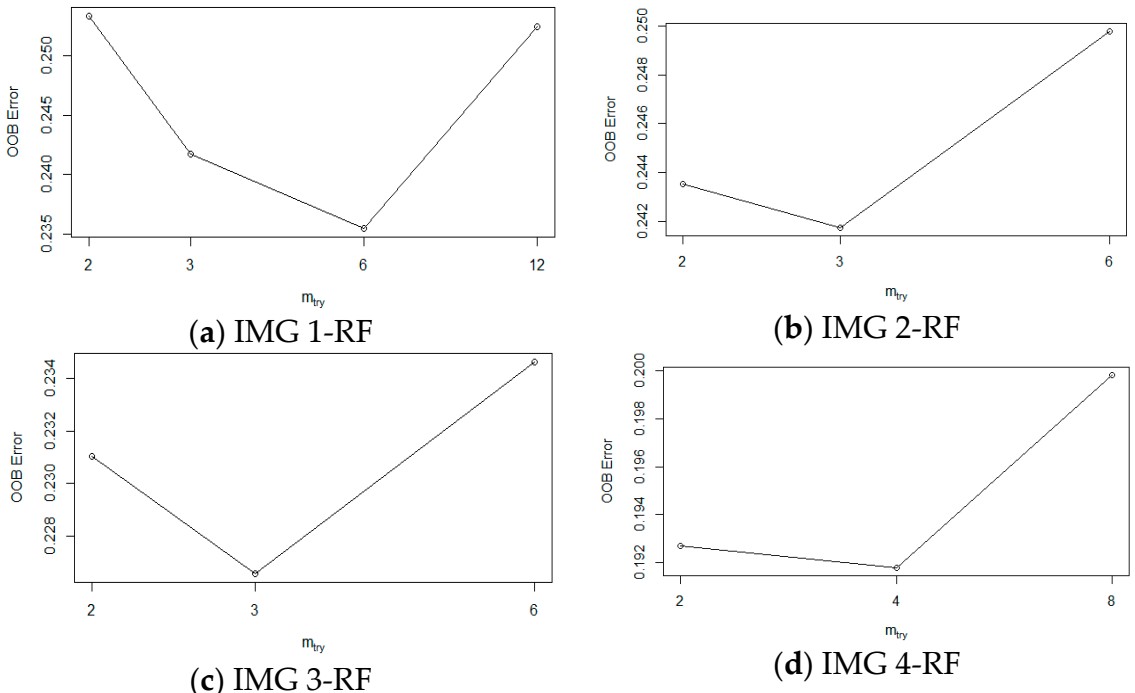

**Figure 6.** The OOB error of the model based on mtry parameter: IMG 1-RF; (**b**) IMG 2-RF; (**c**) IMG 3-RF; (**d**) IMG 4-RF.

*3.2. Analyses*

3.2.1. Accuracy of Classification Results

OA, K, PA, and UA for each class for the different combinations of image groups and classifiers are reported Table 6 and a comparison of the results is reported in Figure 7. In general, relatively large accuracies were found with OA >60% and K >0.6. Of interest, the IMG 4 composite of rainy and dry images produced the greatest accuracies for all classifiers. By contrast, the rainy season IMG 2 images produced the smallest accuracies. Classification accuracies for IMG 1 and IMG 3 were less than for IMG 4, but greater than for IMG 2.

**Table 6.** Accuracy estimates: OA = overall accuracy, K = Kappa, PA = producer's accuracy, UA = user's accuracy, $\hat{A}$ = class area estimate (km$^2$), SE($\hat{A}$) = standard error of class area estimate.

| Image | Classifier | OA | Kappa | Accuracy | Land Class * | | | | | | | | | | |
|---|---|---|---|---|---|---|---|---|---|---|---|---|---|---|---|
| | | | | | 1 | 2 | 3 | 4 | 5 | 6 | 7 | 8 | 9 | 10 | 11 |
| IMG 1 | MLR | 68.3 | 0.657 | PA | 97.60 | 35.50 | 70.00 | 39.40 | 0.000 | 47.70 | 91.40 | 47.10 | 48.60 | 100.00 | 89.30 |
| | | | | UA | 58.50 | 66.70 | 66.70 | 80.00 | 0.000 | 68.80 | 75.90 | 70.00 | 100.00 | 85.70 | 61.50 |
| | | | | $\hat{A}$ | 821.72 | 932.59 | 502.16 | 378.25 | 241.3 | 456.51 | 1923.88 | 502.16 | 202.17 | 91.3 | 469.56 |
| | | | | SE($\hat{A}$) | 104.35 | 156.52 | 97.82 | 97.82 | 84.78 | 110.87 | 195.65 | 123.91 | 45.65 | 13.04 | 71.74 |
| | Ik-NN | 72.1 | 0.732 | PA | 98.20 | 55.40 | 81.90 | 45.10 | 32.50 | 38.30 | 93.60 | 28.20 | 31.60 | 100.00 | 87.70 |
| | | | | UA | 80.00 | 65.20 | 80.80 | 80.00 | 60.00 | 84.60 | 68.60 | 91.70 | 91.70 | 92.30 | 62.10 |
| | | | | $\hat{A}$ | 886.94 | 808.68 | 404.34 | 541.29 | 202.17 | 463.04 | 1878.23 | 456.51 | 189.13 | 97.82 | 593.47 |
| | | | | SE($\hat{A}$) | 78.26 | 143.48 | 78.26 | 123.91 | 71.74 | 130.43 | 208.69 | 123.91 | 52.17 | 6.52 | 104.35 |
| | RF | 67.7 | 0.67 | PA | 92.20 | 64.00 | 62.10 | 42.10 | 46.90 | 18.90 | 89.90 | 38.40 | 42.00 | 100.00 | 87.80 |
| | | | | UA | 77.40 | 63.00 | 85.70 | 81.80 | 66.70 | 100.00 | 59.50 | 85.70 | 91.70 | 92.30 | 58.10 |
| | | | | $\hat{A}$ | 886.94 | 763.03 | 502.16 | 515.21 | 182.61 | 697.81 | 1689.10 | 397.82 | 215.21 | 104.35 | 567.38 |
| | | | | SE($\hat{A}$) | 104.35 | 123.91 | 104.35 | 123.91 | 58.69 | 163.04 | 215.21 | 97.82 | 52.17 | 6.52 | 104.35 |
| | SVM | 73.2 | 0.748 | PA | 94.90 | 54.30 | 73.60 | 47.10 | 26.80 | 36.20 | 94.40 | 42.80 | 46.50 | 100.00 | 84.80 |
| | | | | UA | 76.70 | 64.00 | 87.00 | 100.00 | 60.00 | 78.60 | 70.60 | 92.30 | 92.90 | 92.30 | 63.00 |
| | | | | $\hat{A}$ | 880.42 | 886.94 | 404.34 | 456.51 | 189.13 | 397.82 | 1956.49 | 463.04 | 169.56 | 104.35 | 613.03 |
| | | | | SE($\hat{A}$) | 91.3 | 163.04 | 84.78 | 110.87 | 71.74 | 117.39 | 215.21 | 104.35 | 52.17 | 6.52 | 110.87 |
| IMG 2 | MLR | 63.9 | 0.611 | PA | 67.80 | 37.70 | 71.80 | 33.70 | 8.80 | 96.30 | 85.30 | 39.50 | 41.40 | 100.00 | 86.20 |
| | | | | UA | 54.50 | 58.80 | 66.70 | 53.30 | 10.00 | 84.20 | 67.90 | 70.00 | 90.90 | 92.30 | 64.30 |
| | | | | $\hat{A}$ | 854.33 | 978.24 | 469.56 | 371.73 | 189.13 | 228.26 | 1813.01 | 645.64 | 280.43 | 104.35 | 586.95 |
| | | | | SE($\hat{A}$) | 150.00 | 189.13 | 71.74 | 84.78 | 71.74 | 26.09 | 215.21 | 136.95 | 65.22 | 6.52 | 104.35 |
| | Ik-NN | 64.3 | 0.673 | PA | 90.90 | 36.50 | 61.20 | 44.90 | 56.30 | 42.40 | 84.50 | 38.80 | 38.10 | 86.00 | 82.00 |
| | | | | UA | 74.20 | 80.00 | 62.10 | 85.70 | 83.30 | 85.70 | 51.20 | 81.30 | 92.30 | 91.70 | 63.00 |
| | | | | $\hat{A}$ | 854.33 | 1317.37 | 417.38 | 404.34 | 104.35 | 365.21 | 1643.45 | 436.95 | 182.61 | 91.3 | 704.33 |
| | | | | SE($\hat{A}$) | 104.35 | 202.17 | 91.3 | 84.78 | 39.13 | 110.87 | 228.26 | 110.87 | 58.69 | 13.04 | 123.91 |
| | RF | 67.5 | 0.712 | PA | 86.30 | 39.40 | 65.90 | 66.30 | 53.10 | 58.60 | 85.00 | 42.00 | 28.00 | 100.00 | 80.200 |
| | | | | UA | 78.60 | 68.80 | 75.00 | 91.70 | 62.50 | 87.50 | 58.30 | 100.00 | 91.70 | 85.70 | 56.700 |
| | | | | $\hat{A}$ | 913.03 | 1180.41 | 404.34 | 319.56 | 104.35 | 293.47 | 1760.84 | 547.82 | 195.65 | 91.3 | 717.38 |
| | | | | SE($\hat{A}$) | 110.87 | 202.17 | 84.78 | 45.65 | 39.13 | 84.78 | 228.26 | 117.39 | 65.22 | 13.04 | 136.95 |

**Table 6.** *Cont.*

| Image | Classifier | OA | Kappa | Accuracy | Land Class * | | | | | | | | | | |
|---|---|---|---|---|---|---|---|---|---|---|---|---|---|---|---|
| | | | | | 1 | 2 | 3 | 4 | 5 | 6 | 7 | 8 | 9 | 10 | 11 |
| | SVM | 68.4 | 0.717 | PA | 83.10 | 38.60 | 66.00 | 52.10 | 41.80 | 66.00 | 85.90 | 43.80 | 42.10 | 100.00 | 89.40 |
| | | | | UA | 67.70 | 64.70 | 72.00 | 81.80 | 80.00 | 88.20 | 63.60 | 100.00 | 92.90 | 85.70 | 64.30 |
| | | | | Â | 815.2 | 1180.41 | 436.95 | 345.65 | 130.43 | 247.82 | 1871.7 | 534.77 | 136.95 | 91.3 | 723.9 |
| | | | | SE(Â) | 104.35 | 208.69 | 91.3 | 65.22 | 45.65 | 78.26 | 228.26 | 123.91 | 52.17 | 13.04 | 117.39 |
| IMG 3 | MLR | 64.2 | 0.611 | PA | 73.20 | 36.00 | 69.70 | 29.40 | 6.90 | 96.20 | 84.70 | 50.80 | 48.40 | 100.00 | 85.40 |
| | | | | UA | 54.50 | 58.80 | 66.70 | 53.30 | 10.00 | 84.20 | 67.90 | 70.00 | 90.90 | 92.30 | 64.30 |
| | | | | Â | 939.11 | 971.72 | 430.43 | 384.78 | 221.74 | 254.34 | 1663.01 | 717.38 | 319.56 | 117.39 | 502.16 |
| | | | | SE(Â) | 156.52 | 182.61 | 71.74 | 97.82 | 78.26 | 26.09 | 202.17 | 136.95 | 71.74 | 6.52 | 97.82 |
| | Ik-NN | 66.9 | 0.684 | PA | 88.80 | 40.30 | 87.70 | 67.50 | 38.90 | 18.40 | 90.10 | 38.50 | 37.10 | 86.80 | 87.80 |
| | | | | UA | 75.90 | 60.00 | 69.00 | 78.60 | 100.00 | 54.50 | 60.00 | 81.30 | 85.70 | 91.70 | 70.80 |
| | | | | Â | 919.55 | 965.2 | 319.56 | 280.43 | 169.56 | 723.9 | 1754.32 | 443.47 | 182.61 | 97.82 | 658.68 |
| | | | | SE(Â) | 117.39 | 182.61 | 45.65 | 45.65 | 58.69 | 182.61 | 228.26 | 117.39 | 58.69 | 13.04 | 104.35 |
| | RF | 69.5 | 0.721 | PA | 90.90 | 51.20 | 84.80 | 56.10 | 46.80 | 13.70 | 91.40 | 39.60 | 45.50 | 100.00 | 100.00 |
| | | | | UA | 82.10 | 61.50 | 74.10 | 90.90 | 100.00 | 80.00 | 60.50 | 85.70 | 92.90 | 92.30 | 67.90 |
| | | | | Â | 913.03 | 893.46 | 313.04 | 352.17 | 176.08 | 743.46 | 1754.32 | 502.16 | 150 | 104.35 | 613.03 |
| | | | | SE(Â) | 104.35 | 163.04 | 45.65 | 78.26 | 52.17 | 176.08 | 221.74 | 117.39 | 45.65 | 6.52 | 78.26 |
| | SVM | 71.2 | 0.743 | PA | 92.20 | 42.60 | 86.00 | 68.40 | 50.60 | 25.80 | 94.50 | 37.10 | 43.20 | 88.90 | 97.70 |
| | | | | UA | 77.40 | 59.10 | 83.30 | 92.30 | 100.00 | 85.70 | 65.80 | 81.30 | 100.00 | 91.70 | 66.70 |
| | | | | Â | 886.94 | 1043.46 | 358.69 | 280.43 | 136.95 | 658.68 | 1819.53 | 469.56 | 169.56 | 117.39 | 586.95 |
| | | | | SE(Â) | 104.35 | 189.13 | 45.65 | 39.13 | 45.65 | 150 | 208.69 | 117.39 | 52.17 | 13.04 | 78.26 |
| IMG 4 | MLR | 65.9 | 0.611 | PA | 69.30 | 39.40 | 78.70 | 26.60 | 1.20 | 98.70 | 85.50 | 31.30 | 58.80 | 100.00 | 83.30 |
| | | | | UA | 54.50 | 58.80 | 66.70 | 53.30 | 10.00 | 84.20 | 67.90 | 70.00 | 90.90 | 92.30 | 64.30 |
| | | | | Â | 854.33 | 1030.42 | 560.86 | 345.65 | 195.65 | 456.51 | 1799.97 | 449.99 | 273.91 | 71.74 | 489.12 |
| | | | | SE(Â) | 156.52 | 189.13 | 84.78 | 84.78 | 71.74 | 45.65 | 215.21 | 130.43 | 45.65 | 6.52 | 97.82 |
| | Ik-NN | 74.3 | 0.781 | PA | 91.10 | 52.60 | 69.50 | 44.90 | 39.30 | 55.00 | 94.70 | 35.20 | 47.00 | 100.00 | 87.20 |
| | | | | UA | 75.00 | 81.30 | 90.00 | 92.90 | 57.10 | 87.50 | 67.60 | 93.30 | 92.90 | 92.30 | 70.80 |
| | | | | Â | 867.38 | 1036.94 | 449.99 | 449.99 | 123.91 | 280.43 | 2034.75 | 384.78 | 143.48 | 97.82 | 639.12 |
| | | | | SE(Â) | 110.87 | 176.08 | 84.78 | 130.43 | 52.17 | 84.78 | 228.26 | 104.35 | 45.65 | 6.52 | 104.35 |

**Table 6.** *Cont.*

| Image | Classifier | OA | Kappa | Accuracy | Land Class * | | | | | | | | | | |
|---|---|---|---|---|---|---|---|---|---|---|---|---|---|---|---|
| | | | | | 1 | 2 | 3 | 4 | 5 | 6 | 7 | 8 | 9 | 10 | 11 |
| | RF | 80 | 0.802 | PA | 89.40 | 61.90 | 78.40 | 68.90 | 46.00 | 77.20 | 95.10 | 34.20 | 33.20 | 100.00 | 98.10 |
| | | | | UA | 85.20 | 69.20 | 83.30 | 92.30 | 62.50 | 100.00 | 82.10 | 83.30 | 100.00 | 92.30 | 69.20 |
| | | | | Â | 945.63 | 952.16 | 391.3 | 280.43 | 130.43 | 189.13 | 2204.31 | 482.6 | 176.08 | 91.3 | 678.25 |
| | | | | SE(Â) | 110.87 | 163.04 | 58.69 | 52.17 | 52.17 | 32.61 | 195.65 | 136.95 | 58.69 | 13.04 | 84.78 |
| | SVM | 80.3 | 0.813 | PA | 89.40 | 63.30 | 77.80 | 70.20 | 42.20 | 81.00 | 95.10 | 39.60 | 39.30 | 100.00 | 97.50 |
| | | | | UA | 82.10 | 73.90 | 90.50 | 93.30 | 71.40 | 100.00 | 80.60 | 86.70 | 92.30 | 85.70 | 69.20 |
| | | | | Â | 932.59 | 971.72 | 391.3 | 358.69 | 123.91 | 234.78 | 2223.87 | 710.86 | 378.25 | 97.82 | 639.12 |
| | | | | SE(Â) | 113.48 | 163.04 | 58.69 | 104.35 | 52.17 | 39.13 | 208.69 | 182.61 | 143.48 | 13.04 | 84.78 |

* Class 1: dense evergreen; 2: open evergreen; 3: semi-evergreen; 4: dipterocarp; 5: plantation; 6: rubber; 7: industrial plants; 8: crop land; 9: residential. 10: water surface; 11: other land.

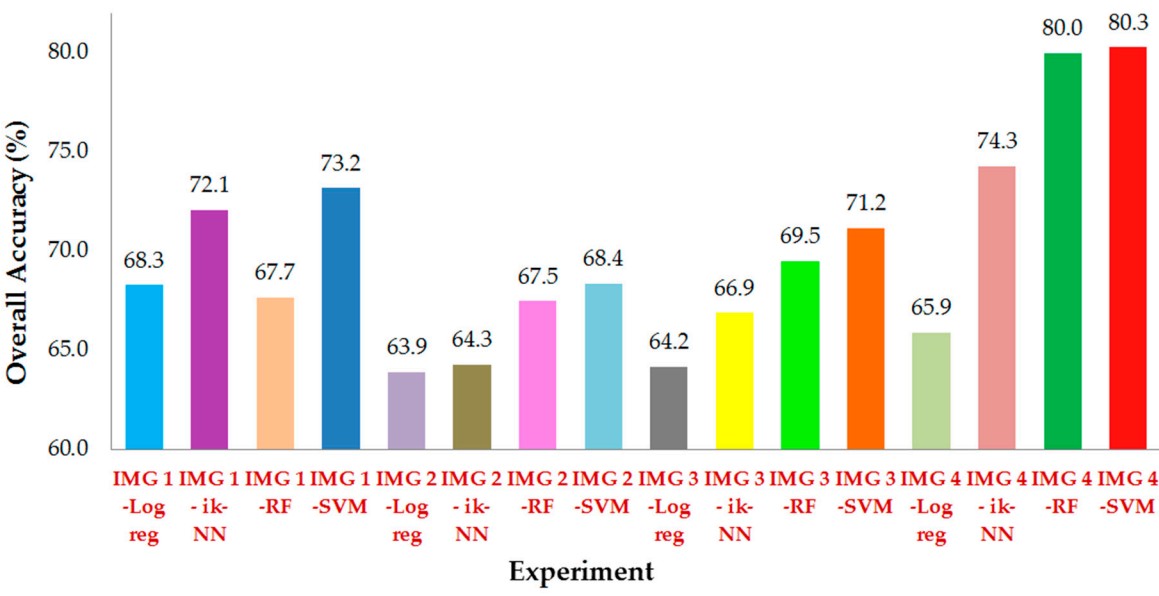

**Figure 7.** Overall accuracy for all combinations.

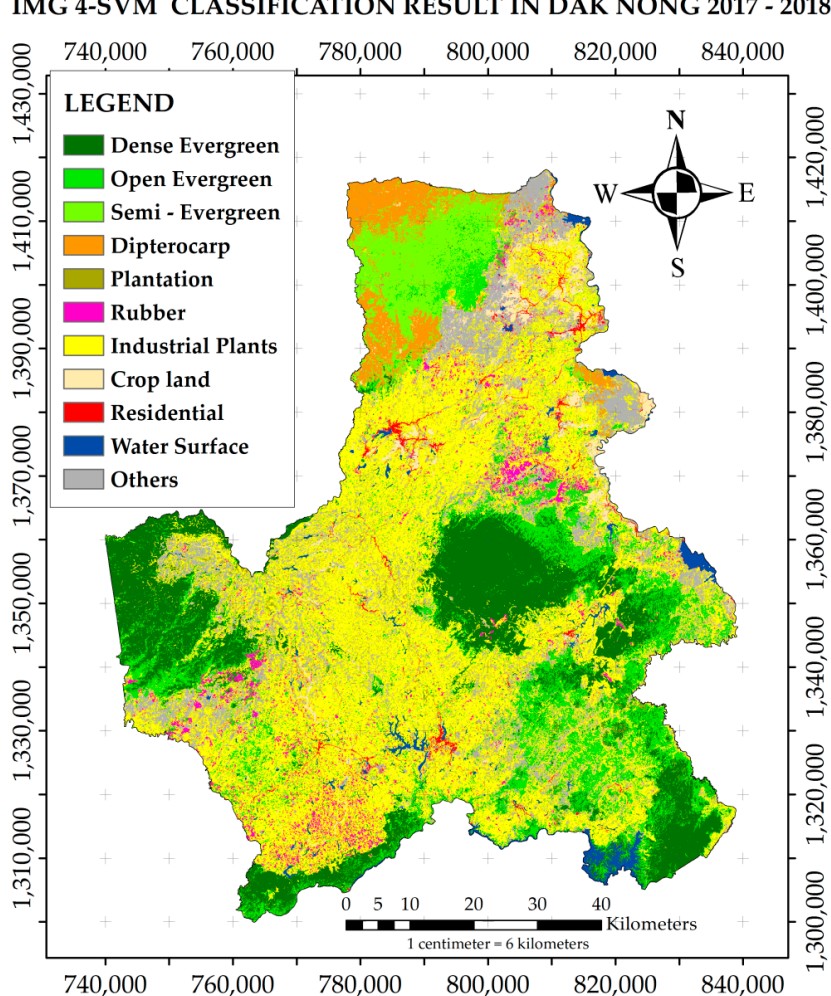

**Figure 8.** Land use and land cover (LULC) map produced for the most accurate classification combination.

The greatest accuracy was achieved for the composite two-season IMG 4 using the SVM classifier, specifically OA = 80.3% and Kappa = 0.813. The smallest accuracy was for IMG 2 with the MLR classifier. The differences between the greatest and smallest accuracies were 16.4% percentage points for OA and 0.202 for K. With respect to the classification algorithms, differences between the greatest and smallest accuracies for the four algorithms were 14.4% percentage points for OA and 0.202 for K. For SVM, the differences were 11.9% percentage points and 0.096 for K; for ik-NN, the differences were 10% percentage points for OA and 0.108 for K. The final LULC map was constructed using the classification for the most accurate IMG 4/SVM combination and is shown in Figure 8.

Average UA and PA estimates were greater than 60%, apart from a few cases, but differed by LULC class. For the forest cover classes, dense forest (1) had the greatest accuracy, while open forest (2) had the smallest accuracies for the four methods for most seasons (Figure 9).

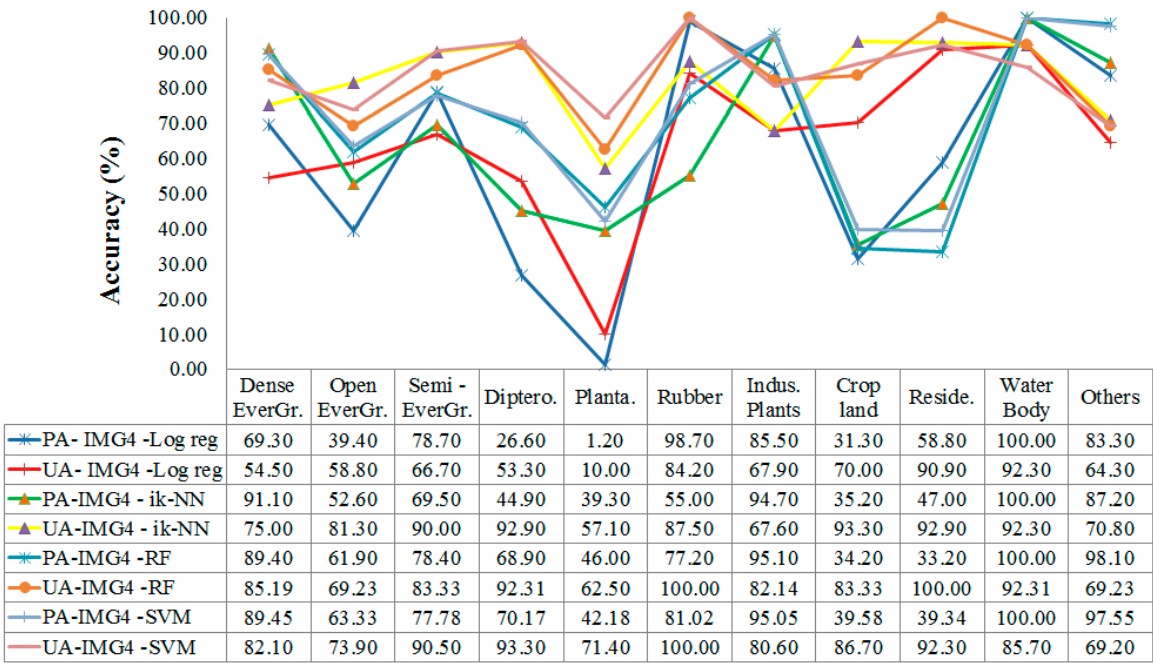

| | Dense EverGr. | Open EverGr. | Semi - EverGr. | Diptero. | Planta. | Rubber | Indus. Plants | Crop land | Reside. | Water Body | Others |
|---|---|---|---|---|---|---|---|---|---|---|---|
| PA- IMG4 -Log reg | 69.30 | 39.40 | 78.70 | 26.60 | 1.20 | 98.70 | 85.50 | 31.30 | 58.80 | 100.00 | 83.30 |
| UA- IMG4 -Log reg | 54.50 | 58.80 | 66.70 | 53.30 | 10.00 | 84.20 | 67.90 | 70.00 | 90.90 | 92.30 | 64.30 |
| PA-IMG4 - ik-NN | 91.10 | 52.60 | 69.50 | 44.90 | 39.30 | 55.00 | 94.70 | 35.20 | 47.00 | 100.00 | 87.20 |
| UA-IMG4 - ik-NN | 75.00 | 81.30 | 90.00 | 92.90 | 57.10 | 87.50 | 67.60 | 93.30 | 92.90 | 92.30 | 70.80 |
| PA-IMG4 -RF | 89.40 | 61.90 | 78.40 | 68.90 | 46.00 | 77.20 | 95.10 | 34.20 | 33.20 | 100.00 | 98.10 |
| UA-IMG4 -RF | 85.19 | 69.23 | 83.33 | 92.31 | 62.50 | 100.00 | 82.14 | 83.33 | 100.00 | 92.31 | 69.23 |
| PA-IMG4 -SVM | 89.45 | 63.33 | 77.78 | 70.17 | 42.18 | 81.02 | 95.05 | 39.58 | 39.34 | 100.00 | 97.55 |
| UA-IMG4 -SVM | 82.10 | 73.90 | 90.50 | 93.30 | 71.40 | 100.00 | 80.60 | 86.70 | 92.30 | 85.70 | 69.20 |

**Figure 9.** User's and producer's accuracies by class for the four classifiers using the IMG 4 combination of the rainy and dry Sentinel 2 satellite imagery.

### 3.2.2. Land Cover Class Area Estimates

Land cover class area estimates with corresponding standard errors are shown by class for the 16 seasonal image and prediction technique combinations in Table 6. Class area estimates ranged from slightly greater than 70 km$^2$ for class 10 for the IMG 4 and MLR combination to slightly greater than 2200 km$^2$ for class 7 for the IMG 4 and SVM combination. Standard errors ranged from approximately 6 km$^2$ to approximately 230 km$^2$, with larger standard errors associated with larger area estimates (Figure 10), although smaller ratios of standard errors to area estimates were associated with larger area estimates. Regardless of the seasonal image and prediction technique combination, the three classes with the greatest areas, in order, were class 7: perennial industrial plants, class 2: open evergreen broadleaved forest, and class 1: dense evergreen broadleaved forest. For IMG 1, IMG 2, and IMG 3, the sums of the estimated areas for these three classes as proportions of the total area ranged from slightly more than 0.50 to approximately 0.63, but with larger estimates for IMG 4.

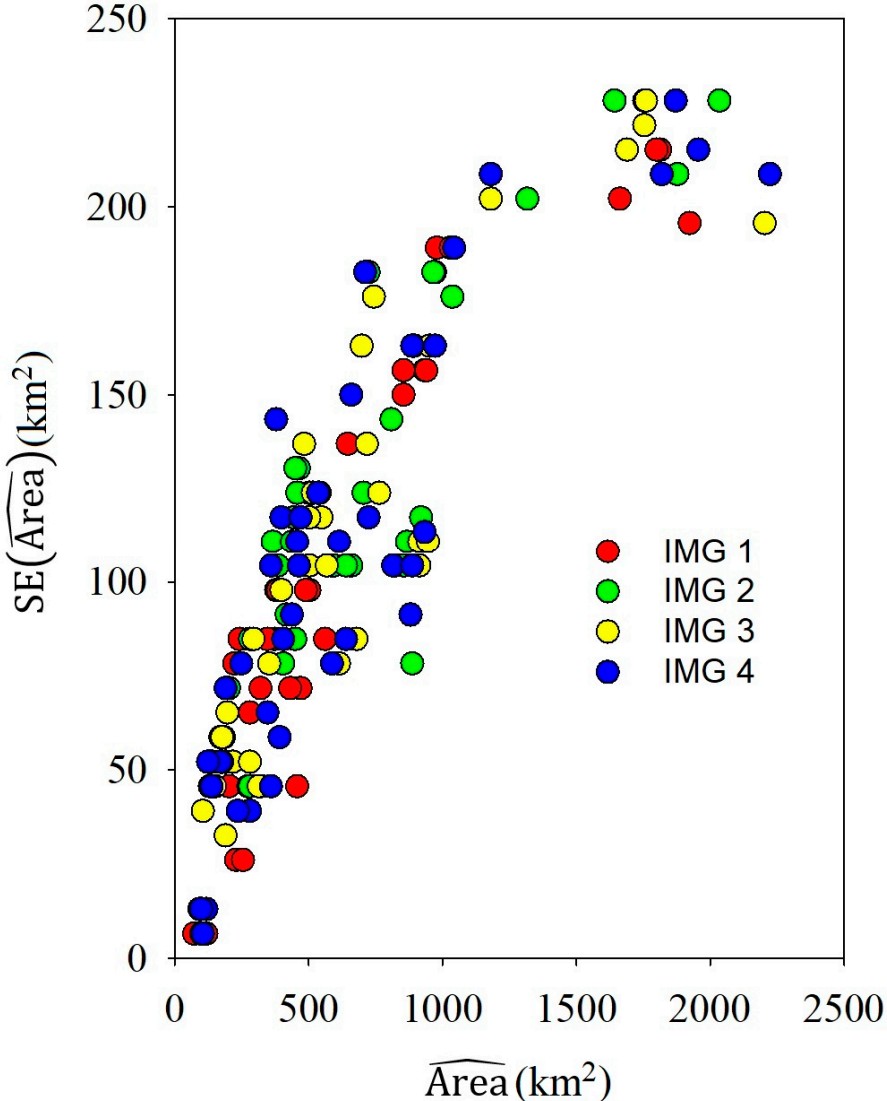

**Figure 10.** Standard errors (SE) (km$^2$) of class area estimates versus area estimates (km$^2$).

## 4. Discussion

Errors are present in any classification, estimation, or prediction [21,86–88]. Comparison of the results of this study and those of earlier studies is not straightforward because the numbers and definitions of the vegetation classes differ by study. Thus, optimality differs by study and user [21,86–88]. There are also no generally accepted limits on how accurate a classification should be to be characterized as reliable, because different users may have different concerns about accuracy. They may, for example, be interested in the accuracy for a specific class or in accuracy for areal estimates [89]. In addition, multiple factors influence classification accuracy: image quality, classifier, image composition, number and details of classes, and sample size.

Andersen et al. (1976) [90] recommended that accuracies of 85% for mapping land cover are acceptable. However, as Foody (2008) [91] noted, for many contemporary mapping applications, the challenge may be more difficult than assessed by Anderson et al. (1976) [90], particularly when attempting to distinguish among a large number of relatively detailed classes at a relatively local, large cartographic scale. Consequently, in such applications, the use of the 85% target suggested by Anderson et al. (1976) [90] may be inappropriate, as it may be unrealistically large.

Indeed, many studies have been conducted to select the most accurate classifier, either among those simultaneously evaluated or with classifiers evaluated in other studies. Such works reach no

consensus, because the performance of a classifier always depends on the specific site characteristics, on the type and quality of the remotely sensed data, and also on the number and general aspects of the classes of interest [13]. Using the RF, SVM, maximum likelihood, and neural network classification algorithms to discriminate among four individual land cover classes based on two Landsat-8 OLI scenes, Lowe and Kulkarni (2015) [40] reported overall classification accuracies of 96.25%, 86.88%, 83.13%, and 76.87%, respectively. Kennedy et al. (2015) [41] used RF to classify Landsat time-series data from 1198 training patches for four classes (agriculture, forest, urbanization, and stream) and reported OA greater than 80%, but most successfully for the numerically well-represented forest management class. Meanwhile, Franco-Lopez (2001) [38] used k-NN to map 13 types of land cover using Landsat TM and achieved OA = 64%. Tomppo et al. (2008) [92] reported OA between 70% and 80% for classifying dominant tree species in one boreal forest test site in Finland when using two adjacent Landsat 7 ETM+ scenes and the ik-NN method. Pelletier et al. (2016) [18] used RF and SVM algorithms to classify SPOT-4 imagery and Landsat-8 HR-SITS images in southern France. The authors reported an OA of 83.3% for RF and 77.1% for SVM. Research by Phan and Kappas (2017) [20] showed different results among RF, SVM, and k-NN classifiers used to discriminate six types of LULC using Sentinel-2 image data in the Red River Delta of Vietnam. This research reported that SVM produced the greatest OA (95.29%) with the least sensitivity to the training sample sizes, followed consecutively by RF (94.44%) and k-NN (94.13%). These results indicate that no standard of accuracy is appropriate for all cases, because accuracy relevance depends on both the objective and the user.

Spatial information including remotely sensed data has been an excellent source of information for decision makers in forest management, albeit in conjunction with an understanding of classification uncertainties, whereby the probabilities of non-optimal and infeasible decisions are reduced. For this study, OA ranged from 63.9% to 80.3% (Figure 7) when using Sentinel 2 data to classify 11 LULC classes, with SVM producing the greatest accuracies. The difference between accuracies for the most accurate SVM classifier and the least accurate MLR classifier was approximately 14.4%. Although the results for SVM and RF were relatively similar, some authors recommend RF because training is less time-consuming and parameter selection is easier [18], a recommendation that was confirmed in our study.

Producer's and user's accuracies among the 11 LULC classes differed considerably (Figure 9). In general, the open evergreen forest classes were confused more than the other forest cover classes. This result is attributed to the heterogeneous conditions of natural tropical forests. In addition, forests in the study area have been disturbed to different degrees [21]. Among the forest classes, deciduous dipterocarp and semi-evergreen forest are considered the most challenging for remote sensing classification because of the seasonal deciduous characteristics of these forest types in the dry season [93]. However, this problem may be solved by using the combination of dry and rainy season images, as investigated in the present study.

The Sentinel-2 images acquired for different seasons (plant growth stages) produced different results. The greatest accuracies were for the composite rainy and dry season IMG 4; by contrast, the lowest accuracies were for the rainy season IMG 2. The observed reflectance varied by season owing to changes in the solar illumination geometry caused by the Earth's translation movement. In addition, the vegetation in the study area varies depending on the season, owing to the substantial rainfall differences for the two seasons. Sothe et al. (2017) [13] assert that differences in classification accuracies for the dry and rainy seasons can be attributed to the differences in solar illumination geometry between the two seasons. For images acquired in the dry season, the incident sun radiation arrives in a more perpendicular direction to the Earth's surface, thus reducing the shadow effect caused by topography and variations in the forest canopy height, and leading to greater pixel illumination. For the current study, there was a substantial increase in classification accuracies when using a composite of dry and rainy Sentinel 2 images (IMG 4). For the ik-NN, RF, and SVM classifiers, the greatest accuracies were obtained for the combined rainy and dry IMG 4 relative to the rainy or dry season alone (Table 6). The accuracy increase for the composite image may be explained by the fact that different seasons

contain different information for the same kind of land cover (e.g., dipterocarp forest is deciduous in dry seasons and green in rainy seasons). Combining the two season's image bands captures additional information on land cover.

Among all combinations of images, classification algorithms, and land classes, the smallest SE for area estimates was for the water surface class owing to its stability, whereas the largest SE was for the industrial plant class. In fact, because cultivation characteristics of industrial plants in the study area are quite complex with a variety of species such as coffee, pepper, and cashew, all with uneven ages, large SEs are inevitable. This complexity also explains the large difference among area estimates for this class, ranging from 1643.45 km$^2$ to 2223.87 km$^2$, or from 25% to 34% of the total area (Figure 10).

Although classification accuracies for vegetation classes were not particularly large, the classifications are still useful for complex tropical rain forests that have been disturbed to different degrees such as in the Central Highlands of Vietnam. The area estimates and spatial distributions of the LULC classes produced from the current study will assist local authorities, managers, and other stakeholders in decision-making and planning regarding forest land cover and uses. The usual practice is for the Institution of Forest inventory and Planning (FIPI) to conduct a forest inventory and construct a forest map every five years. Local forest units such as Dak Nong receive the maps and update them manually. However, the accuracy of the map has usually not been announced, and inaccuracies and errors have been detected only by local forest staff when patrolling in the field. Moreover, LULC changes, particularly for industrial land, occur quickly and easily owing to factors such as unstable crop markets and increasing population resulting from migration. Thus, the results of this study will not only provide authorities with updated information on current conditions, but will also serve as a recommendation regarding methods for proactively updating LULC maps in a timely and costly manner. Specifically, timely and updated maps assist authorities by serving as a basis for formulating suitable solutions and policies for managing LULC including forest cover.

## 5. Conclusions

This research showed the utility of combining Sentinel-2, multi-spectral, and dry and rainy season band data when mapping LULCs in Dak Nong Province, Vietnam. The greatest accuracies were achieved for the composite IMG 4 obtained by combining dry and rainy season image sets using the SVM classifier.

Among the classifiers, SVM produced the greatest accuracies, although RF, which had similar accuracies, was simpler to train and apply, and was less computationally intensive. For IMG 4, the greatest accuracies with SVM were OA = 80.3% and Kappa index = 0.813; for RF, the greatest accuracies were OA = 80.0% and K = 0.802. Thus, the combination of dry and rainy season imagery used with the SVM or RF may contribute to potentially new ways for classifying the complex tropical forest of Vietnam and similar areas. The area estimates and spatial distributions of the LULC classes produced from the current study will assist local authorities, managers, and other stakeholders in decision-making and planning regarding forest land cover and uses.

In conclusion, the two-season, multi-spectral Sentinel-2 images provided useful data for classifying LULC classes in areas with substantial fragmentation, especially for natural forests that have been disturbed and degraded at different levels such as in Dak Nong, Vietnam. The SVM and RF machine learning algorithms were both accurate classifiers when used with the Sentinel 2 imagery. The methods developed for this study are applicable to boreal and temporal forests with different classes in addition to the tropical forests for the current study. However, the model parameters always need to be re-estimated for each application.

**Author Contributions:** Conceptualization, H.T.T.N., E.T., and R.E.M.; methodology, H.T.T.N., E.T., and R.E.M.; software, H.T.T.N., T.M.D., E.T., and R.E.M.; validation, H.T.T.N., E.T., R.E.M., and T.M.D.; formal analysis, H.T.T.N., R.E.M., E.T., and T.M.D.; investigation, T.M.D. and H.T.T.N.; resources, H.T.T.N.; data curation, H.T.T.N. and T.M.D.; writing—original draft preparation, H.T.T.N. and T.M.D.; writing—review and editing, R.E.M. and E.T.; visualization, T.M.D.; supervision, H.T.T.N.; project administration, H.T.T.N.; funding acquisition, H.T.T.N. All authors have read and agreed to the published version of the manuscript.

**Funding:** This research was funded by UNITED STATES AGENCY FOR INTERNATIONAL DEVELOPMENT, grant number AID-OAA-A-11-00012.

**Acknowledgments:** This work is part of the research project under the PEER program (Partnerships for Enhanced Engagement in Research), a U.S. government program to fund scientific research in developing countries. This is a program sponsored by USAID in partnership with several other U.S. Government agencies and administered by the U.S. National Academy of Sciences (NAS). The authors would like to thank all of the people involved in collecting field data for classification and validation. The authors thank also the editor and three anonymous reviewers for the constructive comments.

**Conflicts of Interest:** The authors declare no conflict of interest.

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
