# Peer review of "Land Use/Land Cover Mapping Using Multitemporal Sentinel-2 Imagery and Four Classification Methods—A Case Study from Dak Nong, Vietnam"

_remotesensing, doi:10.3390/rs12091367_

Round 1

Reviewer 1 Report

The manuscript dealt with the Land Use/Land Cover mapping using Sentinel 2 imagery using different classification methods – a case study from Dak Nonng, Vietnam. Their work focused on the selection of datasets from different time intervals and uses them to develop a land cover map using four different algorithms. There is not any innovation in terms of the methodology adopted and the result obtained, however, a comprehensive analysis was done reasonably. The manuscript preparation and result presentation are done poorly which makes it very difficult to follow the paper, before going for the publications I recommend to improve the presentation. The specific comments are as follows: A through re-structuring and improvement in writing is necessary to make the manuscript readable. All figures are extremely poor in quality, some of them are not readable, I recommend to keep the high-resolution figures. The captions of all figures and tables are not enough to understand and read the figures and tables. They should be descriptive enough to understand it. I don’t understand the writing inline 207, 208, 213, they should be formatted properly and proper citation should be done. I don’t understand the way selected the ground points used for training and validation, please make follow some standard method, and should not be biased about the region or land cover types. The proportion of the validation dataset is low, how did you divide this dataset? It should be written clearly. A validation dataset is less than 20% which is too low, you have to use at least 30% of the sample for the validation. Line 268: please follow the guideline for the citation. Part of the content in the results should be in the methodology or not appropriate in this section.

Author Response

A general response

We thank the Editors and the reviewers for handling our manuscript and for providing constructive comments. We have addressed all comments and taken them into account or given rebuttals for comments with which we disagree. Please find the specific responses below.

Reviewer # 1

Comment: (x) Extensive editing of English language and style required 

Response: The paper was extensively edited by a native English writer who has a degree in English and has received awards for editing contributions.  Further, Reviewer #2 complimented us on the clear writing.  Finally, unless the reviewer clarifies or provides some examples of what is unclear, we do not know how to respond to the comment.

Comment: The manuscript dealt with the Land Use/Land Cover mapping using Sentinel 2 imagery using different classification methods – a case study from Dak Nonng, Vietnam. Their work focused on the selection of datasets from different time intervals and uses them to develop a land cover map using four different algorithms. There is not any innovation in terms of the methodology adopted and the result obtained, however, a comprehensive analysis was done reasonably. The manuscript preparation and result presentation are done poorly which makes it very difficult to follow the paper, before going for the publications I recommend to improve the presentation. The specific comments are as follows:

      A through re-structuring and improvement in writing is necessary to make the manuscript readable.

Response: First, regarding innovation, several aspects of the study are relatively unique: 1) we used extensive image data from multiple seasons with top of atmosphere scenes to acquire usable image data, 2) we compared a large number classifiers, 3) we used statistically rigorous estimation and error estimation procedures, not commonly used in remote sensing based LULC classification, and 4) we documented the utility of multi-season data to facilitate estimation for this study area which is characterized by considerable topographic, climatic, phenological, and biophysical variation.

      Second, regarding the presentation, as we have previously noted, the paper has been extensively edited by a native English writer.  In addition, another reviewer complimented us on a well-written paper. 

      Third, we added Section 2.1, Overview, at the beginning of Chapter 2 and a flowchart to help to understand the methods and approach. 

Comment:  All figures are extremely poor in quality, some of them are not readable, I recommend to keep the high-resolution figures. The captions of all figures and tables are not enough to understand and read the figures and tables. They should be descriptive enough to understand it.

Response: The figures were reconstructed and the captions edited.

Comment: I don’t understand the writing inline 207, 208, 213, they should be formatted properly and proper citation should be done. I don’t understand the way selected the ground points used for training and validation, please make follow some standard method, and should not be biased about the region or land cover types.

Response: The text was clarified.

Comment: The proportion of the validation dataset is low, how did you divide this dataset? It should be written clearly. A validation dataset is less than 20% which is too low, you have to use at least 30% of the sample for the validation.

Response: First, for statistically rigorous validation, the validation data must be acquired using a probability sampling design.  Because portion of the data obtained using such a design, our validation data were necessarily limited.  Second, our methodological approach was a form of post-stratified estimation.  For this approach, Särndal et al. (1992, p. 267, 407), whose is an acknowledge international authority, recommends minimum within-stratum sample sizes of 10-20.  For all except one class, we have satisfied the criterion.  This reference is: Särndal, C.-E.; Swensson, B; Wretman, J. Model assisted survey sampling.  Springer, New York, USA, 1992. We added one sentence and the reference.

Comment: Line 268: please follow the guideline for the citation. Part of the content in the results should be in the methodology or not appropriate in this section. 

Response: Descriptions of how optimization of the prediction techniques is done rightfully belongs in the Methods section.  However, because the actual optimization entails a degree of data analysis, selection of values for the optimization parameters represent results and, therefore, belong in the Results section.  Specifically, because Section 3.1 focuses on what we discovered or concluded from optimizing the classifiers. we think that this material belongs to the Results section.

Reviewer 2 Report

Improve the final part of the abstract with the discussion and conclusion of your work.

Maybe, you need to reorganize the section 3.1. it seems methodology and not results.

check the references of the figure, sometimes it is lost: see 442; 515

Authors have to give indications if the approach can produce the same result in different study areas or Extend this part with more reason for your affirmation:

"The methods developed for this study should be expected to be applicable in a straightforward way to boreal and temporal forests with different classes in addition to the tropical forests for the current study"

Author Response

A general response

We thank the Editors and the reviewers for handling our manuscript and for providing constructive comments. We have addressed all comments and taken them into account or given rebuttals for comments with which we disagree. Please find the specific responses below.

Reviewer # 2

Comment:  Improve the final part of the abstract with the discussion and conclusion of your work.

Response:  We enlarged the statements at the end of the Discussion section on how the results of the study will be used by land managers, decision-makers, and policy people. A short statement in the abstract about the area estimates have also been added.

Comment: Maybe, you need to reorganize the section 3.1. It seems methodology and not results.

Response: See our response to the last comment of the Reviewer #1: “Descriptions of how optimization of the prediction techniques is done rightfully belongs in the Methods section.  However, because the actual optimization entails a degree of data analysis, selection of values for the optimization parameters represent results and, therefore, belong in the Results section.  Specifically, because Section 3.1 focuses on what we discovered or concluded from optimizing the classifiers. we think that this material belongs to the Results section.”

Comment:  check the references of the figure, sometimes it is lost: see 442; 515

Response: Thank you. Corrected.

Comment:  Authors have to give indications if the approach can produce the same result in different study areas or extend this part with more reason for your affirmation:

"The methods developed for this study should be expected to be applicable in a straightforward way to boreal and temporal forests with different classes in addition to the tropical forests for the current study"

Response: We slightly modified the current text and added additional text to address this issue.

Reviewer 3 Report

The manuscript titled “Land Use/Land Cover Mapping Using Multitemporal Sentinel-2 Imagery and Four Classification Methods - A case study from Dak Nong, Vietnam” represents interesting original scientific research. The idea of the research is interesting and presents enough novelty. The manuscript should attract an audience in the remote sensing and Land use Land cover mapping at the national scale. The manuscript title is accurate and concise. In the entire manuscript, authors use standard technical and scientific terminology.

After well written and concise Introduction, the authors explained in detail used Materials and developed Methods. Results were conducted according to the scientifically correct approach. The conclusions are logical and based on the results of the research. The paper topics fit in Remote sensing aims and scope, especially in image processing and pattern recognition, as well as remote sensing applications. I recommended this paper to be accepted after minor revisions.

Comments for authors:

  1. Use MDPI standard font (Palatino Linotype) on figures if you can. Increase the quality of the figures.
  2. The variable names must have the same font style and size in equations, on figures, tables, and in the manuscript text. Please describe/introduce all variables used in equations or on figures in the manuscript text.
  3. Is it used fusion/pan-sharpening methods to increase the resolution of 20-m Sentinel-2 bands? Suggest to research and include in the manuscript literature about the topic mentioned above as https://doi.org/10.1016/j.rse.2016.10.030 https://doi.org/10.1080/01431161.2017.1392640
  4. All equations must be adequately cited in the entire paper.
  5. Please, double-check all references and reference style.
  6. Please correct typos and language errors (e.g. page 5, line 189 – “2,3” -> “2, 3”, etc.).
  7. In the author contributions section, please remove the default part of paragraphs, e.g. “For research articles with several authors, a short paragraph specifying their individual contributions must be provided. The following statements should be used and“, etc.

Author Response

A general response

We thank the Editors and the reviewers for handling our manuscript and for providing constructive comments. We have addressed all comments and taken them into account or given rebuttals for comments with which we disagree. Please find the specific responses below.

Reviewer # 3

Comment:  The manuscript titled “Land Use/Land Cover Mapping Using Multitemporal Sentinel-2 Imagery and Four Classification Methods - A case study from Dak Nong, Vietnam” represents interesting original scientific research. The idea of the research is interesting and presents enough novelty. The manuscript should attract an audience in the remote sensing and Land use Land cover mapping at the national scale. The manuscript title is accurate and concise. In the entire manuscript, authors use standard technical and scientific terminology.

      After well written and concise Introduction, the authors explained in detail used Materials and developed Methods. Results were conducted according to the scientifically correct approach. The conclusions are logical and based on the results of the research. The paper topics fit in Remote sensing aims and scope, especially in image processing and pattern recognition, as well as remote sensing applications. I recommended this paper to be accepted after minor revisions.

Response: The authors thank the Reviewer for the encouraging comments.

Comments for authors:

  1. Use MDPI standard font (Palatino Linotype) on figures if you can. Increase the quality of the figures.
  2. The variable names must have the same font style and size in equations, on figures, tables, and in the manuscript text. Please describe/introduce all variables used in equations or on figures in the manuscript text.

Response: Thank you. The MDPI standard is used as well as the variable names in the equations added.

Comment:

  1. Is it used fusion/pan-sharpening methods to increase the resolution of 20-m Sentinel-2 bands? Suggest to research and include in the manuscript literature about the topic mentioned above as https://doi.org/10.1016/j.rse.2016.10.030 https://doi.org/10.1080/01431161.2017.1392640

Response: The re-sampling method is described in Section 2.1.  We have tested fusion methods in resampling with SPOT 5. The tests were not encouraging, even vice versa wherefore we  did not use this it in this study.

  1. All equations must be adequately cited in the entire paper.
  2. Please, double-check all references and reference style.

Response: Thank you. We checked and followed the reference style.

Comment:

  1. Please correct typos and language errors (e.g. page 5, line 189 – “2,3” -> “2, 3”, etc.).

Response: Thank you. Done.

Comment

  1. In the author contributions section, please remove the default part of paragraphs, e.g. “For research articles with several authors, a short paragraph specifying their individual contributions must be provided. The following statements should be used and“, etc.

Response: Thank you. Removed.

Round 2

Reviewer 1 Report

The authors have addressed raised issues, the manuscript is improved significantly and it is almost ready to publish but before publishing, there are few cited references and tables that have errors please check them thoroughly before publications.